# HGT-UCOD: A Hint-Guided Teacher Framework for Unsupervised Camouflaged Object Detection

## Abstract

Camouflaged Object Detection (COD) holds significant potential in various high-stakes applications, yet its progress is fundamentally bottlenecked by a heavy reliance on large-scale, pixel-level annotated data. While Unsupervised Domain Adaptation (UDA) offers a promising path forward, real-world scenarios often impose stricter constraints due to data privacy, leaving us with only a pre-trained source model—a more challenging setting known as source-free domain adaptation. A critical flaw in current methods is their direct use of the source model (e.g., one trained for salient object detection) to generate pseudo-labels. The inherent "saliency bias" of such models—an inclination to find objects that "stand out" rather than "blend in"—results in incomplete and noisy labels that only capture the most conspicuous parts of a target. Self-training on this flawed guidance inevitably falls into confirmation bias, amplifying initial errors and limiting performance. We introduce a paradigm shift in addressing this problem. Instead of treating the biased predictions as mere noise, we innovatively reframe their high-confidence fragments as reliable "hints". Based on this philosophy, we propose HGT-UCOD, a novel Hint-Guided Teacher framework designed to guide the model in inferring the complete object from these sparse yet trustworthy cues. The cornerstone of our framework is a unique teacher pre-adaptation stage. Here, we first cultivate an "expert teacher" by compelling it to learn to infer the full object from partial views containing only these "hints," thus building specialized knowledge. Subsequently, during student refinement, this expert teacher collaborates with the source model to generate high-quality pseudo-labels via a dynamic fusion strategy. This process is further enhanced by strong consistency regularization, which forces the student to learn robust, perturbation-invariant features. To empower this inference, both our teacher and student models are equipped with a novel Dynamic Convolution Mixture (DCM) module, which adaptively generates content-aware kernels to capture the subtle, context-dependent features of camouflaged objects. Extensive experiments on multiple benchmark datasets demonstrate that our method achieves superior performance, establishing a new state-of-the-art for source-free unsupervised COD.

## 1 Introduction

Camouflaged Object Detection (COD) (Fan et al., 2020a) is a critical and highly challenging task in computer vision, focusing on segmenting objects that blend into their surroundings (Price et al., 2019). This technology is crucial for high-stakes domains like automated biodiversity monitoring for detecting well-camouflaged species, high-stakes medical lesion detection in complex scans (Fan et al., 2020b), and enhanced situational awareness in search and rescue operations (Pérez-de la Fuente et al., 2012). While recent methods based on fully-supervised learning have spurred significant progress, their success reveals a fundamental bottleneck: a heavy reliance on large-scale, pixel-level annotated data. This issue is particularly acute for COD, as creating such datasets is not only prohibitively expensive and time-consuming but also requires domain experts to delineate the often-ambiguous boundaries of camouflaged objects, making high-quality data acquisition exceptionally difficult.

To mitigate this dependency on labeled data, Unsupervised Domain Adaptation (UDA) (Liu et al., 2022) offers a promising alternative, aiming to transfer knowledge from a label-rich source domain (e.g., Salient Object Detection, SOD) to an unlabeled target domain (COD). However, in real-world applications and research, we often lack access to the source training dataset and can only utilize a pre-trained source model. This gives rise to a more challenging and practical problem: Source-Free Domain Adaptation (SFDA) (Li et al., 2024). Consequently, our core research question becomes: How can we effectively adapt a pre-trained model to the demanding task of camouflaged object detection, using only the source model itself and unlabeled target images?

Existing SFDA methods face a fundamental conflict when applied to COD. The source models they typically rely on (i.e., SOD models) are trained to find objects that "stand out," whereas the goal of COD is precisely the opposite: to find objects that deliberately "blend in." This inherent conflict of objectives leads to a severe "saliency bias." Specifically, when the source model is used to generate pseudo-labels for camouflaged images, it can only identify the most conspicuous, least "camouflaged" parts of the target, resulting in incomplete and noisy pseudo-labels. Worse still, standard self-training or adaptation methods that naively trust these low-quality labels fall into a vicious cycle of "confirmation bias." The model progressively reinforces the initial errors caused by saliency bias during training, leading it to overfit to misleadingly "salient" fragments and failing to learn the true, complete structure of the camouflaged object. This severely limits its ultimate performance.

We argue that the key to overcoming this dilemma lies in a complete shift in perspective. Instead of treating the biased predictions from the source model as pure noise to be cleaned or filtered, we innovatively reframe their high-confidence regions as reliable "hints." Although these hints are spatially sparse, they represent the model's most certain knowledge about the target. Based on this core insight, we propose a novel "Hint-Guided" learning paradigm. Its motivation is not to passively correct biased pseudo-labels, but to actively use these sparse yet reliable hints to "compel" the model to infer the complete object form. This process forces the model to learn the generalizable, underlying features of camouflage itself (such as subtle differences in contour and texture from the background), rather than merely overfitting to the deceptive signals produced by saliency bias.

To realize this philosophy, we designed HGT-UCOD, a novel framework based on a "Hint-Guided" Teacher-Student paradigm. Our contributions are multi-layered, spanning a comprehensive set of innovations from core ideology to framework design and specific optimization strategies:

- Core Ideological Innovation: We introduce a novel "Hint-Guided" learning paradigm. This paradigm fundamentally reframes the problem, shifting from correcting noisy predictions to inferring complete objects from reliable cues. By repurposing high-confidence regions of biased predictions as trustworthy "hints," our approach actively counters and leverages the "saliency bias" by guiding the model to reason from sparse information.

- Synergistic Framework and Architectural Innovation: We propose a unique framework that operationalizes our paradigm. It features: A Teacher Pre-adaptation stage that cultivates an "expert teacher" specialized in inferring global structure from local hints. A novel Dynamic Convolution Mixture (DCM) module that empowers both teacher and student models. The DCM generates content-adaptive kernels, providing the architectural foundation necessary to capture the subtle, context-dependent patterns of camouflaged objects—a crucial capability for reasoning beyond sparse hints.

- Strategy Optimization Innovation: In the student refinement phase, we devised a sophisticated set of strategic optimizations. This includes a dynamic fusion strategy to adaptively combine knowledge from the expert teacher and the source model for high-quality pseudo-label generation. This is coupled with strong consistency regularization to force the student model to learn robust, perturbation-invariant features, significantly boosting its generalization capabilities.

Extensive experiments on multiple authoritative COD benchmark datasets demonstrate that our method achieves superior performance, with its combined effect surpassing existing unsupervised approaches on most evaluation metrics and establishing a new state-of-the-art (SOTA) for source-free unsupervised camouflaged object detection.

## 2 RELATED WORK

### 2.1 UNSUPERVISED CAMOUFLAGED OBJECT DETECTION

Unsupervised Camouflaged Object Detection (UCOD) (Zhang & Wu, 2023) has emerged as a key research direction to alleviate the heavy reliance on manual annotations. The predominant strategy in this field is UDA, which aims to transfer knowledge from a label-rich auxiliary domain—typically SOD to the unlabeled target COD domain.

Early UCOD research primarily drew upon classical UDA techniques. For instance, Ganin & Lempitsky (2015); Ding et al. (2023) employed adversarial learning with a domain discriminator to encourage the model to learn domain-invariant features, thereby aligning the distributions of the source and target domains within a shared feature space. Other approaches focused on aligning the second-order statistics of feature maps between the two domains.

More recently, the research trend has shifted towards a self-training paradigm, which leverages pseudo-labels generated for target domain images to conduct supervised learning. Methods such as those by Lu et al. (2025); Shou et al. (2025) typically initialize pseudo-labels using predictions from the source SOD model, which are then refined through various strategies. However, these methods are highly susceptible to confirmation bias, where initial errors from the source model are progressively reinforced and amplified during training, ultimately limiting the model's performance.

### 2.2 SOURCE-FREE DOMAIN ADAPTATION

Source-Free Domain Adaptation (SFDA) addresses the practical constraint of source data unavailability during adaptation, leaving only a pre-trained source model and the unlabeled target data. This scenario has spurred the development of innovative techniques that rely solely on the knowledge encapsulated within the source model. A prominent line of work focuses on generating high-quality pseudo-labels for the target data and using them for self-supervision. Pioneering methods like SHOT (Liang et al., 2020) accomplish this through information maximization and by promoting confident, class-separated predictions. Other approaches have explored estimating the quality of pseudo-labels to filter out noise (Kaushik et al., 2024) or leveraging generative models to synthesize features that mimic the source distribution (Chopra et al., 2024).

The Teacher-Student framework has also become a cornerstone of modern SFDA, heavily inspired by its success in semi-supervised learning (Tarvainen & Valpola, 2017). In this paradigm, a "teacher" model provides more stable pseudo-labels to guide the training of a "student" model. The teacher is then updated via the Exponential Moving Average (EMA) of the student's weights, which ensures a stabilizing effect and prevents the model from collapsing into a state of high confidence in its own errors. While these methods have proven effective for tasks like classification (Song & Wang, 2024), their application to the fine-grained, pixel-level task of COD is non-trivial, given that Camouflage objects often lack strong semantic or visual cues, making pseudo-label generation inherently less reliable.

Our work, HGT-UCOD, builds upon this powerful Teacher-Student paradigm but introduces a crucial innovation: a hint-guided teacher pre-adaptation stage. We distinguish our approach by first enabling the teacher to learn robust object-centric priors from sparse yet reliable hints before it guides the student. This core strategy, combined with a novel pseudo-labeling mechanism and a dynamic architectural component designed for capturing camouflaged patterns, enables our method to effectively tackle the unique challenges of UCOD.

## 3 METHOD

### 3.1 OVERVIEW

As illustrated in Fig.1 and inspired by UCOS-DA Zhang & Wu (2023) and UCOD-DPL Yan et al. (2025), our method comprises three core components: a source model, a teacher model, and a student model, which are trained through a two-stage process for unsupervised binary segmentation. The first stage pre-adapts the teacher to learn discriminative features representations. To achieve this, a Difference Perception Module adaptively selects target regions based on the prediction uncertainty

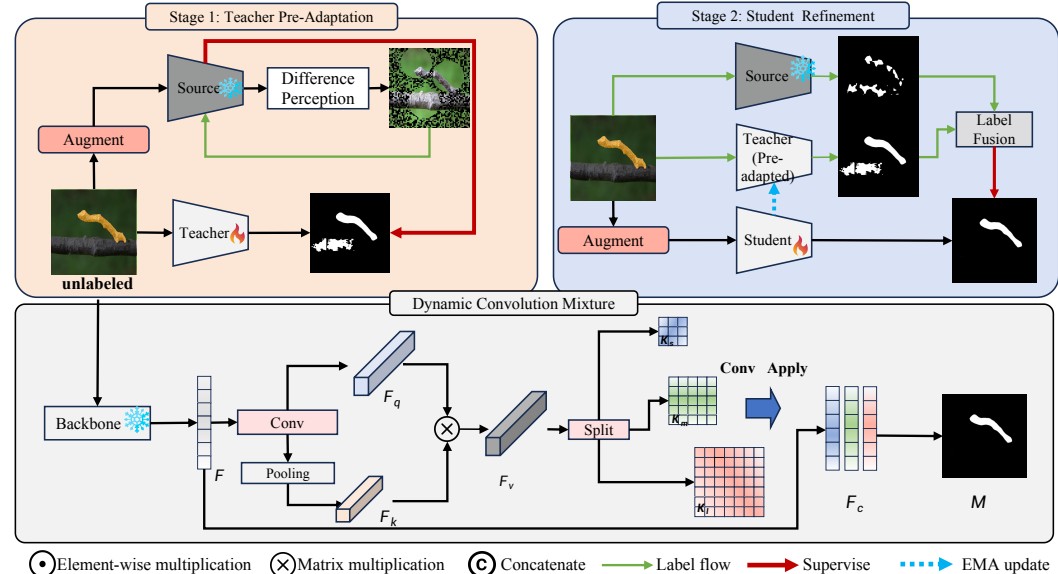

Figure 1: The pipeline of our proposed HGT-UCOD framework. It consists of two core stages: Pre-Adaption and Refinement. The bottom panel details the architecture of our custom Dynamic Convolution Mixture (DCM) module.

of the source model, compelling the teacher model to capture distinct feature paradigms within these areas and consequently produce divergent predictions from the source. In the second stage, the source and teacher models are jointly utilized to generate high-quality pseudo-labels for supervising the student model's training. Both the teacher and student models are built upon an architecture incorporating our novel Dynamic Convolution Mixture (DCM) module. This module is specifically designed to tackle the core challenge of COD by synergizing local feature extraction with dynamic, global context modeling. Its ability to generate content-adaptive kernels is vital for inferring the full object shape from the sparse cues learned during pre-adaptation and for delivering highly refined final predictions.

## 3.2 TEACHER MODEL PRE-ADAPTATION

Before the main training of the student model commences, a crucial preliminary step is the pre-adaptation of the teacher model. The goal of this stage is not merely to replicate the source model, but to cultivate a more specialized instructor whose capabilities are complementary to the pre-trained source model. As illustrated in Fig.1, we employ a specialized training strategy that guides the teacher model to learn to infer an object's complete form from partial yet highly reliable information, inspired by the idea in Chen et al. (2024).

**Difference Perception and Hint Region Generation**: We posit that regions with stable predictions under the perturbations of strong data augmentation represent the most confident parts of the object as identified by the source model, $M_f$. To locate these high-confidence areas, we first compute the discrepancy between the foreground prediction probability map from the original image, $P_f^{orig} = M_f(I)$, and the augmented image, $P_f^{aug} = M_f(\text{Aug}(I))$, where $\text{Aug}(\cdot)$ is the augmentation function. Specifically, this function is composed of a series of transformations, including random rotation, horizontal flipping, Gaussian blur, color jittering, and random cropping. These techniques are combined to create significant visual perturbations, ensuring that only the most structurally stable and confident regions of the object yield consistent predictions. The difference map, $D$, is calculated as:

$$D = \text{Dist}(P_f^{orig}, P_f^{aug}), \tag{1}$$

where $\text{Dist}(\cdot, \cdot)$ is a distance metric. Lower values in $D$ correspond to higher confidence.

To generate K spatially diverse hint points, we employ an iterative greedy selection strategy. First, we select the point $p_1$ with the global minimum value in the difference map D as our initial hint. Then, for each subsequent point $p_{i+1}$ (where $i$ ranges from 1 to $K-1$), we select the point with the lowest D value from all candidate locations that lie outside an exclusion zone of radius $r$ from all previously selected points $\{p_1, \ldots, p_i\}$. This sequential selection is governed by the equation:

$$p_{i+1} = \underset{p \in \mathcal{P}, \, \forall j \leq i, \, \|p - p_j\|_2 > r}{\arg\min} D(p). \tag{2}$$

Here, $\mathcal{P}$ represents the set of all pixel coordinates, and the condition $\|p - p_j\|_2 > r$ ensures that the new point $p_{i+1}$ is at least a distance $r$ from every previously selected point $p_j$. To maintain consistency in our ablation studies on K, we fix $r = 100$. This iterative process guarantees that our high-confidence cues are spatially distributed across the target.

Finally, centered on these $K$ selected points, we generate $K$ fixed-size patches. The union of these patches forms the final binary Hint Mask $M_{hint}$. Applying this mask to the original image $I$ yields the "adaptive hint image," $I_h$:

$$I_h = I \odot M_{hint}, \tag{3}$$

where $\odot$ denotes element-wise multiplication.

The core of this training process lies in its unique supervision mechanism: the teacher model $M_t$ receives the original image $I$ as input, but its learning target is not a ground-truth label. Instead, it is supervised by the source model $M_f$'s prediction on the "adaptive hint image" $I_h$, denoted as $P_{\text{sup}} = M_f(I_h)$. This asymmetric training encourages a functional divergence: while $M_f$ preserves its global sensitivity to salient features, $M_t$ is compelled to specialize in subtle, previously overlooked discrepancies within the constrained yet informative hint regions.

## 3.3 STUDENT MODEL REFINEMENT

Once the teacher model has been pre-adapted to develop its specialized ability to infer the full extent of camouflaged objects from partial cues, we proceed to the training phase of our final prediction model: the student. The core challenge at this stage is how to intelligently fuse the global, generalized knowledge of the source model ($M_f$) with the localized, specialized knowledge of the teacher model ($M_t$) to provide high-quality supervision for the student model ($M_s$).

Inspired by recent works (Yan et al., 2025), we move beyond using a single model for guidance and instead design a dynamic pseudo-label generation strategy. This strategy adaptively adjusts the instructional weights of the source and teacher models based on both the training progress and the inter-model consistency. The central idea is that in the early stages of training, the source model's generalized predictions are more reliable. However, as training progresses, the teacher model—which continuously absorbs knowledge from the student via an EMA—becomes increasingly expert and trustworthy.

Specifically, we dynamically compute a weight $W_{mix}$ at each training step:

$$W_{mix} = (\frac{t}{T} + \|P_f, P_t\|)/2, \tag{4}$$

where $t, T$ denote the current and total number of epochs, $P_f$ and $P_t$ are the predictions from the source and teacher models for the original image $I$. The term $\|...\|$ calculated as the Mean Absolute Error to quantify their overall inconsistency.

Next, we use it to blend the predictions from both models, generating a final soft pseudo-label $P_{gt}$:

$$P_{gt} = W_{mix} \cdot P_t + (1 - W_{mix}) \cdot P_f. \tag{5}$$

**Consistency-Regularized Training:** During training, the student model $M_s$ receives a strongly augmented image $I_e = Aug(I)$ and is tasked with making its prediction $P_s$ align with the dynamic pseudo-label $P_{gt}$ generated on the original, un-augmented image $I$. This consistency learning strategy between strongly and weakly augmented views, inspired by similar methodologies (Lai et al., 2024; He et al., 2023), compels the student model to learn robust features that are invariant to visual perturbations. This significantly enhances its generalization ability and final performance. Concurrently, the student's weights are smoothly transferred to the teacher model via EMA, creating a closed-loop system of continuous self-improvement.

| Methods | CHAMELEON (87) | | | | | CAMO-Test (250) | | | | | COD10K-Test (2026) | | | | | NC4K (4121) | | | | |
|---|---|---|---|---|---|---|---|---|---|---|---|---|---|---|---|---|---|---|---|---|
| | $S_m \uparrow$ | $F_\beta^w \uparrow$ | $F_\beta^m \uparrow$ | $E_\phi \uparrow$ | $M \downarrow$ | $S_m \uparrow$ | $F_\beta^w \uparrow$ | $F_\beta^m \uparrow$ | $E_\phi \uparrow$ | $M \downarrow$ | $S_m \uparrow$ | $F_\beta^w \uparrow$ | $F_\beta^m \uparrow$ | $E_\phi \uparrow$ | $M \downarrow$ | $S_m \uparrow$ | $F_\beta^w \uparrow$ | $F_\beta^m \uparrow$ | $E_\phi \uparrow$ | $M \downarrow$ |
| **Fully-supervised** | | | | | | | | | | | | | | | | | | | | |
| SINet$_{21}$(Fan et al., 2020a) | .872 | .806 | .827 | .946 | .034 | .751 | .606 | .675 | .771 | .100 | .771 | .551 | .634 | .806 | .051 | .808 | .723 | .769 | .871 | .058 |
| UGTR$_{21}$(Yang et al., 2021) | .887 | .794 | .819 | .940 | .031 | .784 | .684 | .735 | .851 | .086 | .817 | .666 | .711 | .890 | .036 | .839 | .746 | .787 | .899 | .052 |
| ZoomNet$_{22}$(Pang et al., 2022) | .902 | .845 | .864 | .958 | .023 | .820 | .752 | .794 | .878 | .066 | .838 | .729 | .766 | .888 | .029 | .853 | .784 | .818 | .896 | .043 |
| HitNet$_{23}$(Hu et al., 2023) | .921 | .897 | .900 | .972 | .019 | .849 | .809 | .831 | .906 | .055 | .871 | .806 | .823 | .935 | .023 | .875 | .834 | .853 | .926 | .037 |
| FSPNet$_{23}$(Huang et al., 2023) | .908 | .851 | .867 | .965 | .023 | .856 | .799 | .830 | .899 | .050 | .851 | .735 | .769 | .895 | .026 | .879 | .816 | .843 | .915 | .035 |
| ZoomNeXt$_{24}$(Pang et al., 2024) | .924 | .885 | .896 | .975 | .018 | .888 | .859 | .875 | .945 | .041 | .898 | .838 | .857 | .955 | .017 | .900 | .865 | .884 | .949 | .028 |
| BiRefNet$_{24}$(Zheng et al., 2024) | **.929** | **.911** | **.922** | .968 | **.016** | **.904** | | .904 | .954 | **.030** | .913 | .874 | .888 | .960 | .014 | .914 | .894 | .909 | .953 | .023 |
| **Semi/Weakly-supervised** | | | | | | | | | | | | | | | | | | | | |
| CRNet$_{23}$(He et al., 2023) | **.818** | **.744** | - | **.897** | .046 | .735 | .641 | - | .815 | .092 | .733 | .576 | - | .832 | .049 | .775 | .688 | - | .855 | .063 |
| PCOD$_{24}$(Chen et al., 2024) | - | - | - | - | - | **.798** | **.727** | - | **.872** | **.074** | **.784** | **.650** | - | **.859** | **.042** | **.822** | **.748** | - | **.889** | **.051** |
| CamoTeacher$_{24}$(Lai et al., 2024) | .756 | .617 | .684 | .813 | .065 | .701 | .560 | .742 | .795 | .112 | .759 | .594 | .836 | .854 | .049 | .791 | .687 | .842 | .868 | .068 |
| **Unsupervised** | | | | | | | | | | | | | | | | | | | | |
| BigGW$_{21}$(Voynov et al., 2021) | .547 | .244 | .294 | .527 | .257 | .565 | .299 | .349 | .528 | .282 | .528 | .185 | .246 | .497 | .261 | .608 | .319 | .391 | .565 | .246 |
| TokenCut$_{22}$(Wang et al., 2022) | .654 | .496 | .536 | .740 | .132 | .633 | .498 | .543 | .706 | .163 | .658 | .469 | .502 | .735 | .103 | .725 | .615 | .649 | .802 | .101 |
| TokenCut$_{22}$w/$B.S.$(Wang et al., 2022) | .655 | .351 | .393 | .582 | .169 | .639 | .383 | .434 | .595 | .195 | .666 | .334 | .399 | .609 | .127 | .735 | .478 | .547 | .683 | .133 |
| SelfMask$_{22}$(Shin et al., 2022) | .617 | .483 | .536 | .698 | .176 | .637 | .431 | .469 | .679 | .131 | .716 | .593 | .634 | .777 | .114 | | | | | |
| SelfMask$_{22}$w/$U.B.$(Shin et al., 2022) | .629 | .447 | .491 | .683 | .169 | .627 | .495 | .547 | .708 | .182 | .645 | .440 | .478 | .687 | .125 | .723 | .601 | .642 | .784 | .110 |
| FOUND$_{23}$(Siméoni et al., 2023) | .684 | .542 | .590 | .810 | .095 | .685 | .584 | .633 | .782 | .129 | .670 | .482 | .520 | .751 | .085 | .741 | .637 | .674 | .824 | .084 |
| FOUND$_{23}$ * (Siméoni et al., 2023) | .832 | .761 | .789 | .915 | .038 | .780 | .715 | .751 | .861 | .086 | .764 | .638 | .665 | .843 | .048 | .812 | .749 | .779 | .887 | .055 |
| UCOD-DA$_{23}$(Zhang & Wu, 2023) | .715 | .591 | .629 | .802 | .095 | .701 | .606 | .646 | .784 | .127 | .689 | .513 | .546 | .740 | .086 | .755 | .656 | .689 | .819 | .085 |
| UCOD-DPL$_{25}$(Yan et al., 2025) | .864 | **.825** | .838 | **.931** | .031 | .793 | .747 | .779 | .862 | .077 | .834 | .763 | .779 | **.916** | **.031** | .850 | .818 | .835 | .923 | **.043** |
| EASE$_{25}$(Du et al., 2025) | .819 | .741 | - | .899 | .044 | .749 | .684 | - | .831 | .098 | .866 | .656 | - | .773 | .040 | .800 | .735 | - | .884 | .056 |
| Ours | **.869** | .815 | **.843** | **.938** | **.033** | **.813** | **.761** | **.791** | **.877** | .075 | **.839** | **.766** | **.785** | .913 | .034 | **.858** | **.827** | **.843** | **.929** | .044 |

Table 1: Performance comparison of state-of-the-art models on CAMO, COD10K, and NC4K datasets. The best results are highlighted in Bold, and the second-best are underlined. * indicates a version reimplemented by us.

## 3.4 Dynamic Convolution Mixture

To effectively reason about the complete form of a camouflaged object from sparse "hints", a model requires a powerful ability to perceive subtle, context-dependent patterns. Standard convolutions with static kernels are ill-suited for this task. Therefore, we equip both our teacher and student models with a novel Dynamic Convolution Mixture (DCM) module, the architecture of which is detailed in Fig.1. The core design philosophy of the DCM is to synergize the strength of traditional convolution in extracting local, static patterns with the capability of attention-like mechanisms in modeling long-range, dynamic contextual dependencies. Its objective is to generate a feature representation that adaptively adjusts to each specific input image, thereby providing robust support for the final, fine-grained segmentation task.

**Dynamic Kernel Generation**: The generation of dynamic convolutional kernels is pivotal for the module to achieve content awareness. This process is designed to dynamically convolutional kernels for each spatial location in the image based on the global information of the input features.

Initially, the input feature map $F$ is passed through a lightweight $1 \times 1$ convolutional layer and pooling layer. Its output is subsequently split into two parallel feature branches, yielding feature maps $F_l$ and $F_s$.

Then, We compute the affinity between $F_q$ and $F_k$ via Matrix Multiply. The core objective of this operation is to aggregate global context and generate a dynamic feature $F_v$. Unlike a simple attention map, $F_v$ encodes parameters for constructing convolutional kernels, enabling subsequent convolutions to move beyond static weights and adapt effectively to image content. The above content can be expressed as:

$$\begin{cases} F_l & = \phi_l(F), F_s = \phi_s(F) \\ F_k & = F_l \bigotimes F_s \end{cases},$$

(6)

where $\phi$ denotes a convolutional layer and $\bigotimes$ represents matrix multiplication.

**Multi-scale Dynamic Convolution Application**: To generate multi-scale convolutional kernels, the context tensor $F_k$ is first passed through a projection layer $\phi_{proj}$. The resulting output is then split into three independent sets of parameters to form three distinct kernels: $K_s$, $K_m$, and $K_l$. This approach facilitates multi-scale feature extraction within a single module. By deriving three distinct kernel sets from the same context-aware feature, the model learns to apply its dynamic context across different scales with high parameter efficiency. This process can be formally expressed as:

$$K_s, K_m, K_l = \text{Split}(\phi_{proj}(F_k)).$$

(7)

Figure 2: Visual example comparison: Comparison of our method with some of the previous state-of-the-art methods. Our method significantly outperforms other methods in capturing the contour details of camouflaged objects, and does not suffer from the fragmentation prediction seen in other methods.

The dynamic convolution operation with a kernel $K_a$ applied to the feature map $F$ at spatial location $(i, j)$ and output channel $c$ can be expressed as:

$$(F * K_a)(i, j, c) = \sum_u \sum_v \sum_d K_a(u, v, d, c) \, F(i + u, j + v, d), \tag{8}$$

where the summations are over the spatial support of the kernel and the input channels $d$. Here, $K_a(u, v, d, c)$ denotes the dynamically generated kernel weight.

The three dynamically generated kernels $\{K_s, K_m, K_l\}$ are applied in parallel, yielding three feature maps that are concatenated along the channel dimension and fused by a standard convolution layer to produce the final mask:

$$M(i, j) = \text{Conv}\big(\big[(F * K_s)(i, j, :), \, (F * K_m)(i, j, :), \, (F * K_l)(i, j, :)\big]\big). \tag{9}$$

### 3.5 Loss Function

Our optimization strategy is systematically organized into two distinct training stages, each employing a tailored loss function. Initially, in the Teacher Pre-adaptation stage, the objective is to cultivate a specialized teacher model. This is achieved by using a Binary Cross-Entropy (BCE) loss:

$$\mathcal{L}_{\text{adapt}} = \mathcal{L}_{\text{BCE}}(P_t, P_{sup}), \tag{10}$$

where the teacher's prediction ($P_t$) is supervised by the prediction of the frozen source model on a "hint image" ($P_f$).

Subsequently, in the main Student Model Refinement stage, the student model is optimized using a composite objective:

$$\mathcal{L}_{\text{total}} = \mathcal{L}_{\text{BCE}}(P_s, P_{\text{gt}}) + \mathcal{L}_{L1}(P_s, P_t), \tag{11}$$

This loss function jointly performs two tasks: the BCE term aligns the student's output ($P_s$) with a dynamically fused pseudo-groundtruth ($P_{gt}$), while the L1 term enforces structural consistency by directly distilling knowledge from the refined teacher's prediction ($P_t$) to the student. This two-phase approach first creates an expert teacher and then leverages it to guide the student with both probabilistic and structural constraints.

## 4 EXPERIMENTS

### 4.1 EXPERIMENTS SETTINGS

**Datasets**: We utilize the identical test dataset employed in prior works (Yin et al., 2024; Fan et al., 2021; Shou et al., 2025), employing a composite training dataset comprising 1,000 images from the CAMO-Training (Le et al., 2019) subset and 3,040 images from the COD10K-Training (Fan et al., 2020a) subset. Following standard unsupervised learning protocols, no ground-truth labels were utilized during training. For comprehensive evaluation, we test our model on three established benchmark datasets: CAMO-Test, COD10K-Test, and NC4K (Lv et al., 2021), collectively representing diverse challenging scenarios in camouflaged object segmentation.

**Evaluation Metrics**: Consistent with established practices in the field, our evaluation employs five principal metrics: the Structure measure ($S_m$) (Fan et al., 2017), the weighted F-measure ($F_\beta^w$) (Margolin et al., 2014),the mean F-measure($F_\beta^m$), the mean E-measure ($E_\phi$) (Fan et al., 2018), and the mean absolute error ($M$) (Perazzi et al., 2012). These metrics are universally adopted in COD literature, enabling fair benchmarking across the CAMO, COD10K, and NC4K test sets.

**Implementation Details**: Our framework is implemented using PyTorch, with training and inference tasks distributed across four NVIDIA A800 GPUs. We adopt DINOv2 as the backbone encoder, leveraging its powerful unsupervised visual representation capabilities to ensure rich spatial feature extraction. Additionally, we employ FOUND (Siméoni et al., 2023) as our source model. All input images are resized to 518×518. Optimization uses the AdamW algorithm with a learning rate of 3e-4, weight decay of 2e-3, and a batch size of 32 per GPU. The model is trained for 30 epochs.

### 4.2 COMPARISON WITH STATE-OF-THE-ARTS

**Quantitative Evaluation**:In Tab.1, we compared our proposed method's performance with competing USOD and UCOD models on three COD test datasets. The results show that our model outperformed all existing USOD and UCOD methods across all metrics and datasets, thus achieving State-Of-The-Art performance. Additionally, our model has surpassed several semi-supervised and fully-supervised methods across all datasets, demonstrating its superior performance, effectiveness, and robustness.

**Visual Comparison**: We present a visual comparison with several SOTA approaches in Fig.2. As illustrated, our model consistently generates more coherent and complete masks, exhibiting a significant advantage in capturing the intricate contour details of camouflaged objects that are often missed or fragmented by competing methods. These results compellingly substantiate the robustness and precision of our final predictions. Furthermore, Fig.3 illustrates the internal refinement process via heatmaps. This visualization reveals that while the baseline source model's predictions are noisy, our hint-guided pre-adaptation produces a much cleaner

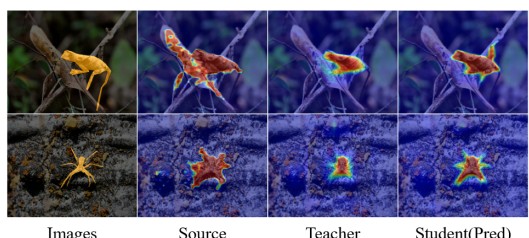

| Images | Source | Teacher | Student(Pred) |

Figure 3: Visualization of the progressive prediction refinement.

and more focused Teacher. The final Student refines these robust cues to recover fine-grained details, clearly demonstrating the effectiveness of our progressive adaptation strategy.

### 4.3 ABLATION STUDY

**Effectiveness of Overall Module:** To validate each key component, we conducted a comprehensive ablation study, with the results summarized in Tab.2. Our analysis follows a progressive integration, starting from a baseline model and systematically adding each proposed module. The results demonstrate a clear synergistic effect: the Teacher-Student (Tea-Stu) paradigm first establishes a robust foundation, which is then significantly amplified by our core contribution, the hint-guided Teacher Pre-adaptation stage. Subsequently, the Dynamic Convolution Mixture (DCM) module further enhances the model's adaptive capacity, while Consistency-Regularized Training (CRT) stabilizes the

| Settings | | | | CAMO-Test (250) | | | | | COD10K-Test (2026) | | | | | NC4K (4121) | | | | |
|---|---|---|---|---|---|---|---|---|---|---|---|---|---|---|---|---|---|---|
| Tea-Stu | Pre-adaption | DCM | CRT | $S_m$↑ | $F_\beta^w$↑ | $F_\beta^m$↑ | $E_\phi$↑ | $M$↓ | $S_m$↑ | $F_\beta^w$↑ | $F_\beta^m$↑ | $E_\phi$↑ | $M$↓ | $S_m$↑ | $F_\beta^w$↑ | $F_\beta^m$↑ | $E_\phi$↑ | $M$↓ |
| | Baseline | | | .703 | .635 | .671 | .781 | .101 | .684 | .584 | .611 | .753 | .088 | .722 | .613 | .645 | .775 | .084 |
| ✓ | | | | .753 | .684 | .721 | .807 | .094 | .749 | .676 | .685 | .813 | .074 | .788 | .717 | .743 | .829 | .062 |
| ✓ | ✓ | | | .773 | .714 | .740 | .821 | .090 | .771 | .695 | .728 | .853 | .049 | .801 | .743 | .774 | .849 | .044 |
| ✓ | | ✓ | | .800 | .747 | .766 | .854 | .086 | .815 | .721 | .752 | .873 | .042 | .832 | .761 | .797 | .901 | .053 |
| ✓ | ✓ | ✓ | | .807 | .754 | .772 | .862 | .082 | .821 | .735 | .763 | .891 | .039 | .841 | .787 | .813 | .910 | .048 |
| ✓ | ✓ | | ✓ | .793 | .731 | .752 | .851 | .090 | .802 | .721 | .751 | .889 | .045 | .831 | .793 | .810 | .904 | .056 |
| ✓ | | ✓ | ✓ | .804 | .749 | .774 | .867 | .083 | .818 | .749 | .763 | .897 | .040 | .843 | .805 | .825 | .918 | .050 |
| ✓ | ✓ | ✓ | ✓ | **.813** | **.761** | **.791** | **.877** | **.075** | **.839** | **.766** | **.785** | **.913** | **.034** | **.858** | **.827** | **.843** | **.929** | **.044** |

Table 2: Ablation study on the components of our framework. We retrained our model with different settings on the same learning rate and epochs.

process and boosts generalization. The consistent performance gain at each step validates that every component plays an indispensable role in achieving our final state-of-the-art results.

| $k$ | CAMO-Test (250) | | | | COD10K-Test (2026) | | | | $w$ | CAMO-Test (250) | | | | COD10K-Test (2026) | | | |
|---|---|---|---|---|---|---|---|---|---|---|---|---|---|---|---|---|---|
| | $S_m$↑ | $F_\beta^m$↑ | $E_\phi$↑ | $M$↓ | $S_m$↑ | $F_\beta^m$↑ | $E_\phi$↑ | $M$↓ | | $S_m$↑ | $F_\beta^m$↑ | $E_\phi$↑ | $M$↓ | $S_m$↑ | $F_\beta^m$↑ | $E_\phi$↑ | $M$↓ |
| 1 | .804 | .777 | .862 | .079 | .814 | .778 | .904 | .038 | 5 | .808 | .781 | .865 | .080 | .831 | .765 | .901 | .037 |
| 2 | .809 | .789 | .871 | .078 | .821 | .781 | .908 | .036 | 10 | **.813** | **.791** | **.877** | **.075** | **.839** | **.785** | **.913** | **.034** |
| 3 | **.813** | **.791** | .877 | **.075** | **.839** | .785 | **.913** | **.034** | 15 | .804 | .774 | .873 | .081 | .834 | .768 | .908 | .038 |
| 4 | .810 | .784 | **.880** | .077 | .834 | **.787** | .911 | .036 | 20 | .793 | .763 | .865 | .085 | .819 | .758 | .885 | .042 |

Table 3: Ablation studies on the number of selected hint points ($k$, left) and the number of Pre-adaptation epochs ($w$, right). We highlight the best-performing values.

**Hyperparameters of Difference Perception and Hint Region Generation:** In our confidence-guided point selection strategy, the number of hint points, $k$, and the number of pre-adaptation epochs, $w$, are two critical hyperparameters. To determine their optimal values, we conducted a detailed ablation study. As shown in Tab.3, we evaluated the model's performance on the CAMO and COD10K datasets while varying both $k$ and $w$. The results for $w$ indicate that model performance reached its zenith at $w = 10$. A shorter duration appears insufficient for the teacher model to fully adapt to the target domain, whereas a longer duration leads to a significant performance degradation. This decline is attributed to the model overfitting to the sparse hint regions.

**Ablation Study on Hint Region Generation Strategy:** To validate the principle of our "hint-guided" approach, we conducted an ablation study on the hint generation strategy, comparing our high-confidence guided method with fully random masking and low-confidence guided masking. As shown in Tab.4, the random strategy performed worst, indicating that unguided masking leads to spurious learning.

| Methods | CAMO-Test (250) | | | COD10K-Test (2026) | | |
|---|---|---|---|---|---|---|
| | $S_m$↑ | $F_\beta^w$↑ | $E_\phi$↑ | $S_m$↑ | $F_\beta^w$↑ | $E_\phi$↑ |
| high-confidence | **.813** | **.761** | **.877** | **.839** | **.766** | **.913** |
| random | .801 | .745 | .852 | .814 | .732 | .883 |
| low-confidence | .807 | .751 | .858 | .823 | .748 | .891 |

Table 4: Ablation study on different hint generation strategies.

The low-confidence strategy was also suboptimal, as focusing on ambiguous regions without reliable anchors hindered coherent object representation. In contrast, our method, leveraging stable high-confidence regions, provided a robust foundation, highlighting that hint reliability is the key factor for effective adaptation.

## 5 CONCLUSION

In this work, we addressed the critical challenge of source data inaccessibility in Unsupervised COD by proposing HGT-UCOD, a novel teacher framework centered on a "hint-guided" learning philosophy. The cornerstone of our approach is a unique teacher pre-adaptation stage, where a confidence-guided strategy compels the teacher to infer complete objects from sparse yet highly reliable hints, effectively creating an expert guide. During student refinement, this expert knowledge is synergized with the source model to generate high-quality pseudo-labels, a process enhanced by our Dynamic Convolution Mixture module for adaptively capturing complex structures. The entire framework is further stabilized by a consistency learning scheme, which significantly boosts model generalization and robustness. Extensive experiments on multiple benchmark datasets demonstrate that HGT-UCOD sets a new SOTA, proving that by first teaching a model to master what it reliably knows, we can effectively guide it to perceive the unknown.

## 6 ETHICS STATEMENT

This work adheres to the ICLR Code of Ethics. In this study, no human subjects or animal experimentation was involved. All datasets were sourced in compliance with relevant usage guidelines, ensuring no violation of privacy. We have taken care to avoid any biases or discriminatory outcomes in our research process. No personally identifiable information was used, and no experiments were conducted that could raise privacy or security concerns. We are committed to maintaining transparency and integrity throughout the research process.

## 7 REPRODUCIBILITY STATEMENT

We have made every effort to ensure that the results presented in this paper are reproducible. All codes are available at https://anonymous.4open.science/r/HGT-UCOD_anonymous/README.md to facilitate replication and verification. The experimental setup, including training steps, model configurations, and hardware details, is described in detail in the paper. Additionally, the datasets mentioned in this paper are publicly available, ensuring consistent and reproducible evaluation results.We believe these measures will enable other researchers to reproduce our work and further advance the field.

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

# A  APPENDIX

## A.1  THE USE OF LARGE LANGUAGE MODELS

Large Language Models (LLMs) were used to aid in the writing and polishing of the manuscript. Specifically, we used an LLM to assist in refining the language, improving readability, and ensuring clarity in various sections of the paper. The model helped with tasks such as sentence rephrasing, grammar checking, and enhancing the overall flow of the text.

Table 5: Quantitative analysis of pseudo-label quality and model evolution on the COD10K dataset.

| Model Variant | $S_m \uparrow$ | $F_\beta^w \uparrow$ | $F_\beta^m \uparrow$ | $E_\phi \uparrow$ | $M \downarrow$ |
|---|---|---|---|---|---|
| Source | .764 | .638 | .665 | .843 | .048 |
| Teacher (warmup) | .733 | .613 | .702 | .773 | .051 |
| **Student (Ours)** | **.839** | **.766** | **.785** | **.913** | **.034** |
| Teacher (final) | .812 | .758 | .769 | .900 | .037 |

# B  ADDITIONAL EXPERIMENTAL RESULTS AND ANALYSIS

## B.1  QUALITY ANALYSIS OF PSEUDO-LABELS

To quantitatively validate the effectiveness of our "Hint-Guided" paradigm, we evaluated the quality of the pseudo-labels generated at different stages of our framework. Table 5 presents the performance metrics on the COD10K dataset for the Source model, the Pre-adapted Teacher (after Stage 1), the Final Student, and the Final Teacher (after Stage 2).

## B.2  COMPUTATIONAL COMPLEXITY AND EFFICIENCY

Table 6: Comparison of model configurations and complexity.

| Method | Input Size | Backbone | Parameters (M) | |
|---|---|---|---|---|
| | | | Fixed (Backbone) | Learnable |
| TokenCut | $480 \times 480$ | DINOv1-S | $\approx 21$ | - |
| SelfMask | $224 \times 224$ | DINOv1-S | $\approx 21$ | - |
| FOUND | $224 \times 224$ | DINOv1-S | $\approx 21$ | - |
| UCOD-DA | $512 \times 512$ | DINOv1-B | $\approx 85$ | - |
| FOUND* | $518 \times 518$ | DINOv2-L | $\approx 300$ | - |
| UCOD-DPL | $518 \times 518$ | DINOv2-L | $\approx 300$ | - |
| EASE | $476 \times 476$ | DINOv2-L | $\approx 300$ | - |
| **Ours** | $518 \times 518$ | DINOv2-L | $\approx 300$ | **16** |

To assess the practical efficiency of HGT-UCOD, we summarize the architectural configurations and computational costs in Table 6. Backbone and Model Size. Following the trend of recent state-of-the-art methods, our framework adopts the powerful DINOv2-L as the backbone to ensure robust feature extraction. While the backbone introduces a substantial parameter count ($\approx$300M), it is important to note that these parameters are pre-trained and largely shared. The specific learnable parameters introduced by our method amount to only 16M. Notably, our method achieves efficient training completion within merely 1 hour while maintaining a real-time inference speed of 9.8 FPS. This indicates that our architectural innovations yield significant performance gains with minimal additional parameter overhead.

## B.3  SIZE-AWARE PERFORMANCE ANALYSIS

Table 7: Performance Across Different Sizes

| | COD10K(2026) | SMALL(1379) | MEDIUM(609) | LARGE(38) |
|---|---|---|---|---|
| $S_m$ | 839 | .828 | .863 | .829 |
| $F_\beta^w$ | 766 | .728 | .843 | .889 |
| $F_\beta^m$ | 785 | .734 | .891 | .929 |
| $E_\phi$ | 913 | .902 | .936 | .895 |
| $M$ | 034 | .030 | .042 | .065 |

To investigate the robustness of our model across different object scales, we partitioned the COD10K test set into three groups based on the object area ratio: Small ($< 10\%$), Medium ($10\% - 40\%$), and Large ($> 40\%$).

As shown in Tab. 7, our method performs robustly across objects of different scales. In particular, it achieves outstanding results on large camouflaged objects, which often exhibit internal texture inconsistency and ambiguous boundaries that challenge conventional segmentation approaches. The proposed DCM module, especially its large-kernel branch, effectively models long-range contextual dependencies, thereby preserving structural coherence and improving segmentation quality for large-scale targets.

### B.4 VISUALIZATION AND INTERPRETABILITY

We provide additional qualitative results to intuitively explain the working mechanism of HGT-UCOD.

**Dynamic Perception of DCM.**

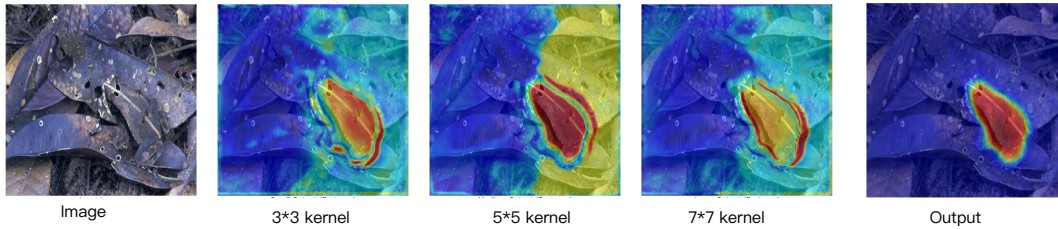

Figure 4: Visualization of DCM modules with different kernel sizes

To understand how the DCM module handles complex camouflage, we visualize the attention weights and output features of its three internal branches in Fig.4 .

- The Small Kernel branch (3×3) focuses on high-frequency details, effectively capturing object boundaries and intricate textures.
- The Large Kernel branch (7×7) exhibits a smoother attention distribution, capturing long-range context and the semantic body of the camouflaged object.
- The Medium Kernel branch (5*5)acts as a bridge, capturing local object parts.

The complementary nature of these branches allows the model to dynamically perceive both fine-grained details and global shapes.

**Evolution from Hints to Whole.** Fig.5 provides a visual explanation of why our "Hint-Guided"

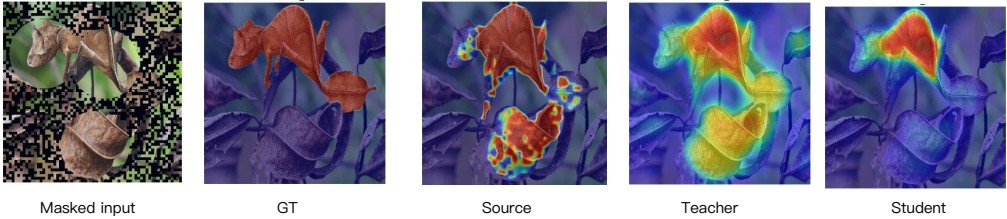

Figure 5: Visualization of the Hint-Guided Pre-adaptation mechanism under occlusion.

strategy works. During the pre-adaptation stage, we intentionally corrupt the input image with random masking and noise (shown as "Masked input"), simulating a scenario where key visual cues are missing or abnormal. Crucially, the Source Model's prediction on this input is highly fragmented and noisy. However, the Teacher Model, having been forced to infer the whole from the parts, successfully reconstructs the complete shape of the leaf-tailed gecko despite the occlusion. This "reconstruction-from-hints" capability is then distilled into the final Student Model, resulting in detection as shown in the last column.

**Multi-object Hint Distribution.**

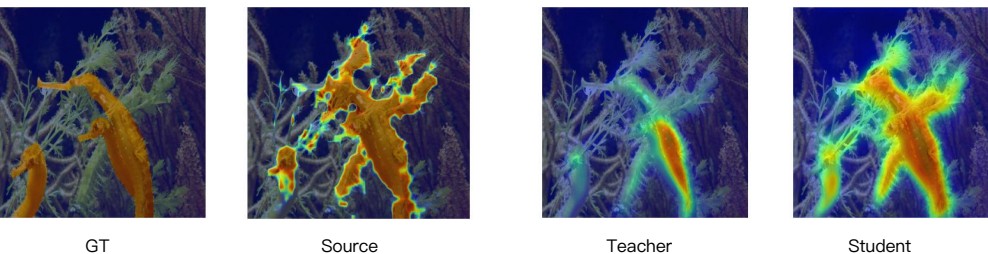

Figure 6: Qualitative evaluation in a complex multi-object scenario.

To address concerns regarding the model's ability to handle scenes with multiple camouflaged instances, we present a visual example in Fig.6. The Source Model fails to separate the targets from the background, generating excessive noise. In contrast, our Student Model, guided by the spatially distributed hints, successfully suppresses the background distractors and segments the individual object. This demonstrates that our hint generation strategy effectively prevents the model from collapsing onto a single salient point.

