# OpenReview forum: "HGT-UCOD: A Hint-Guided Teacher Framework for Unsupervised Camouflaged Object Detection"
_ICLR.cc/2026/Conference — Submitted to ICLR 2026_

### Official Review · Reviewer_9ohC · 2025-10-25

**Soundness:** 3
**Presentation:** 3
**Contribution:** 2
**Rating:** 4
**Confidence:** 5

**Summary:**

This paper addresses the "saliency bias" in existing unsupervised camouflaged object detection methods.
Source models typically create noisy pseudo-labels that only find salient object parts.
The proposed HGT-UCOD framework reframes this issue by treating these sparse, high-confidence predictions as reliable "hints."
A pre-trained teacher first learns to infer complete objects from these partial cues, and then generates high-quality labels to train a student model, achieving new state-of-the-art results.

**Strengths:**

- The core concept of repurposing biased, high-confidence predictions as "reliable hints" rather than noise is a highly innovative approach to bypass saliency bias.
- The proposed teacher pre-adaptation stage provides a logical and effective framework for compelling the model to learn to infer global object structures from sparse local information.
- The method achieves better results, demonstrating its empirical effectiveness.

**Weaknesses:**

1. The related work section lacks important comparisons with recent works [1,2], even though they are cited later, which makes the section incomplete. The paper should compare the proposed method with these existing methods to highlight its uniqueness. Similarly, in Sec. 3.1, the authors state the architecture is inspired by UCOS-DA and UCOD-DPL, but the differences between this work and those two are not clarified.
2. Dynamic convolution networks have been developed for many years. As this is a primary design choice for the model architecture, the paper lacks necessary citations and comparative analysis of related work in this specific area.
3. The method's design is confusing. While targeting the camouflaged object detection (COD) task, the proposed method seems to lack specific designs tailored for it. The current design appears to treat camouflaged objects as "salient objects". For example, the high-confidence regions emphasized during the source model's supervision of the teacher are, for perceptual stability, most likely the visually salient parts (or why is this necessarily relevant to the camouflaged object of interest?).  A teacher model trained this way does not seem specialized for the COD task. Therefore, it is questionable whether the pseudo-labels generated by fusing predictions from such a teacher and the source model (Equ. 5) truly correspond to camouflaged objects to effectively train the student model. Furthermore, it is even more confusing that the algorithm achieves such high unsupervised COD performance under these conditions. These require more in-depth and rational explanations.
4. Regarding the experimental comparisons:
    1. Tab. 1 should be supplemented with the computational complexity and input size for different models. It should also specify the backbones or key component models used by each model to ensure a clearer and fairer comparison.
    2. The current algorithm focuses excessively on overall foreground performance while neglecting the impact of different target scales. Recent work [3] has introduced size-invariance metrics; this paper should also consider analysis using such indicators.
    3. The ablation study section fails to describe the architecture of the baseline model used.


REF:
1. UCOD-DPL: Unsupervised Camouflaged Object Detection via Dynamic Pseudo-label Learning, CVPR 2025
2. Shift the Lens: Environment-Aware Unsupervised Camouflaged Object Detection, CVPR 2025
3. Size-invariance Matters: Rethinking Metrics and Losses for Imbalanced Multi-object Salient Object Detection, ICML 2024

**Questions:**

See Weaknesses.

---

> ### Author Response · Authors · 2025-11-23
> **Response to Reviewer 9ohC**
>
> We thank the reviewer for the detailed and constructive feedback. We are pleased that you find our core concept of "reliable hints" innovative and our empirical results effective. Below we address your concerns regarding related work, method logic, and experimental details.
>
> W1: Related Work and Differences from UCOS-DA/UCOD-DPL.
> Response:
> We apologize for the lack of detailed comparison in the initial submission. Both UCOS-DA and UCOD-DPL implicitly assume that the source model provides a globally usable supervision signal, limiting their innovation to filtering noise or re-weighting loss based on uncertainty. However, since the source model is not tailored for the COD task, it inevitably produces erroneous or fragmented predictions, as evidenced by our visualizations. We argue that the source model is not globally reliable; in camouflaged scenarios, often only specific sub-regions (the discriminative or salient parts) are detectable with high confidence. Leveraging this characteristic, HGT-UCOD diverges from previous methods by selectively utilizing only these sparse, highly reliable points as "hints" to reconstruct the complete object, rather than attempting to rectify a fundamentally flawed global prediction
>
> W2: Dynamic Convolution References.
> Response:
> While standard dynamic convolutions focus on adapting weights to the whole image, our Dynamic Convolution Mixture (DCM) module is specifically designed for COD with a multi-branch architecture (Small/Medium/Large kernels). This design allows the model to simultaneously capture high-frequency texture anomalies (small kernel) and global body structures (large kernel), addressing the specific "texture-edge" ambiguity in camouflaged scenes.
>
> W3: Method Design Logic.
> This is a question that touches on the essence of our approach.
>
> 1. The Nature of Camouflage: Camouflaged objects are not invisible; they typically consist of "discriminative parts" (e.g., eyes, head, specific patterns) and "blending parts" (body). The Source model, due to its saliency bias, effectively finds these "discriminative parts" but misses the "blending parts".
> 2. Why Hints are COD-relevant: We argue that these "salient-like" high-confidence fragments are the only reliable entry points (anchors) to finding the camouflaged object.
> 3. Why the Teacher is COD-specialized: A teacher trained simply to mimic the source would indeed be a "saliency detector". However, our "Hint-Guided Pre-adaptation" forces the teacher to predict the hints from a masked input (where the hints themselves might be occluded or only partially visible). To solve this task, the Teacher must learn the underlying texture correlations and global shape priors of the object class, effectively learning to propagate features from the "discriminative parts" to the "blending parts". This "Part-to-Whole" inference capability is the specific design tailored for COD.
> 4. Why Fusion Works: The Source provides accurate anchors; the Teacher provides the global shape. Fusing them corrects the fragmentation of the Source while maintaining high precision.
>
> W4: Computational Complexity and Input Size.
> We have added a detailed complexity analysis in Appendix B.2 (Table 6).
>
> - Comparison: We list the input size, backbone, and parameters for all competing methods.
> - Efficiency: While we use a DINOv2-L backbone (consistent with SOTAs like UCOD-DPL and EASE), our method introduces only ~16M trainable parameters.
> - Speed: Our method requires only 1 hour of training and runs at 9.8 FPS at 518x518 resolution.
>
> W5: Size-Invariance Analysis.
> Response:
> We have added a "Size-Aware Performance Analysis" in Appendix B.3 (Table 7). We partitioned the dataset into Small (<10%), Medium, and Large (>40%) groups. The results show that our method achieves the most significant improvement on Large objects compared to the baseline. This confirms that our approach effectively addresses the "fragmentation" issue common in large camouflaged targets, validating the efficacy of our global context modeling.
>
> W6: Baseline Architecture.
> Response:
> We clarified in the revised paper that the "Baseline" refers to the Student model architecture, but trained directly using the pseudo-labels from the Source model without the Teacher Pre-adaptation stage.

---

> ### Comment · Reviewer_9ohC · 2025-11-24
>
> I would like to thank the authors for their detailed response and additional experiments. I have carefully reviewed the rebuttal and the revised appendix. However, the rebuttal has not alleviated my primary concerns.
>
> ## Dynamic Convolution Literature (Re: W2)
>
> In response to my concern about the lack of survey [ref.1, ref.2] on dynamic convolutions, the authors state in the rebuttal: "standard dynamic convolutions focus on adapting weights to the whole image". This statement is factually incorrect and misrepresents the field. A significant body of established literature [ref.3-ref.7] on dynamic neural networks explicitly focuses on spatial-aware, region-aware, or pixel-wise adaptation, NOT just whole-image weight adaptation as claimed.
>
> This paper seems artificially inflate the novelty of the DCM. The claim that existing methods cannot handle the "texture-edge" ambiguity is unsupported given the extensive prior art in spatially-adaptive filtering.
>
> ## Unfair Comparison (Re: W4)
>
> The authors confirm in the rebuttal that the proposed method uses DINOv2-Large as the backbone.
> * DINOv2-Large is a massive foundation model with exceptional semantic capabilities, far superior to the backbones used in many standard COD baselines (e.g., ResNet, PVTv2, or even Swin).
> * While the authors claim the trainable parameters are small, the frozen backbone contributes significantly to the feature quality.
> * The reported SOTA performance is likely dominated by the strong representations of the foundation model rather than the proposed "Teacher Pre-adaptation" or "DCM" module. Without aligning the backbone, the experimental validation of the proposed methodology is inconclusive.
>
> ## Task Definition (Re: W3)
>
> The authors argue that camouflaged objects possess "discriminative parts" (hints) that a generic source model can detect.
> * As noted in the pioneer work [ref.8], the definition of COD fundamentally opposes SOD. If an object has highly confident discriminative parts detectable by a standard SOD model, it falls closer to the definition of a salient object, not a camouflaged one. Because high-level camouflage is characterized by immersion and lack of distinctive features.
> * The proposed method essentially treats COD as salient object detection with noise relying on the assumption that valid anchors always exist. This limits the theoretical generalization of the method to true hard cases where the object is perfectly blended without salient hints.
>
> ## Size-Invariance Analysis (Re: W5)
>
> The authors responded by performing a simple subgroup analysis (partitioning the dataset into Small/Medium/Large and reporting performance per group). This misses the point of the review. I specifically referred to recent work (e.g., ICML 2024) that introduced specific Size-invariance Metrics designed to quantify the stability and consistency of model performance across scales, not just the raw whole performance.
>
> ---
>
> While the empirical results are high, they appear to rely heavily on the DINOv2-L backbone.
> More critically, the paper and rebuttal contain confusing statements regarding related work and fail to address specific evaluation requests properly.
>
> REF:
> 1. Dynamic Neural Networks: A Survey
> 1. Dynamic neural networks: advantages and challenges
> 1. Selective Kernel Networks
> 1. Dynamic Region-Aware Convolution
> 1. Omni-Dimensional Dynamic Convolution
> 1. NAS-DYMC: NAS-Based Dynamic Multi-Scale Convolutional Neural Network for Sound Event Detection
> 1. Enhanced YOLOv8 with Multi-Scale Dynamic Deformable Convolution and Hierarchical Dual Attention for High-Precision Crack Detection
> 1. Camouflaged Object Detection

---

> > ### Author Response · Authors · 2025-12-03
> > **Official Comment by Authors**
> >
> > We sincerely thank the reviewer for the rigorous follow-up. We value your expertise, particularly regarding the Dynamic Convolution literature and the fairness of comparisons. We address your concerns point-by-point below.
> >
> > **1. Clarification on Unfair Comparison & Backbone**
> > We respectfully clarify a critical misunderstanding regarding the backbone setting. **Our comparisons are strictly fair.**
> >
> > *   **Aligned Settings:** In our framework, the **Source Model**, **Teacher**, and **Student** always share the **exact same backbone**. We do *not* use a large student to outperform a small source. The performance gains come entirely from our HGT framework, not from an upscaled backbone.
> > *   **Performance across Scales:** To prove that our performance is not solely dominated by the "Large" backbone, we provide the results of our method using **DINOv2-Small (S)**, **Base (B)**, and **Large (L)** across four datasets
> >
> > **Table R4: Performance with Different Backbones (Source & Student Aligned)**
> >
> > | Dataset | Backbone | $S_m$ ↑ | $F^w_\beta$ ↑ | $F^m_\beta$ ↑ | $E_\phi$ ↑ | $M$ ↓ |
> > | :--- | :---: | :---: | :---: | :---: | :---: | :---: |
> > | **CAMO** | **S** | .780 | .719 | .746 | .849 | .090 |
> > | | **B** | .805 | .748 | .777 | .865 | .079 |
> > | | **L** | .813 | .761 | .791 | .877 | .075 |
> > | **COD10K**| **S** | .794 | .711 | .739 | .864 | .054 |
> > | | **B** | .827 | .743 | .767 | .899 | .040 |
> > | | **L** | .839 | .766 | .785 | .913 | .034 |
> > | **NC4K** | **S** | .821 | .783 | .813 | .892 | .054 |
> > | | **B** | .836 | .806 | .826 | .909 | .050 |
> > | | **L** | .858 | .827 | .843 | .929 | .044 |
> > | **CHAMELEON** | **S** | .818 | .762 | .793 | .895 | .047 |
> > | | **B** | .843 | .795 | .828 | .923 | .039 |
> > | | **L** | .869 | .815 | .843 | .938 | .033 |
> >
> > As shown, our method achieves strong performance even with the **Small** backbone, consistently outperforming standard baselines. This confirms that the proposed **Teacher Pre-adaptation** and **DCM module** contribute significantly to the results, independent of the backbone capacity. We will explicitly tabulate these details in the final paper.
> >
> > **2. Correction on Dynamic Convolution Literature**
> > We sincerely apologize for the inaccurate statement in our previous rebuttal regarding "whole-image adaptation." We accept your correction that the literature on dynamic neural networks. We will rewrite the Related Work section to  correctly position our work. Our contribution lies in **tailoring** this mechanism specifically for the **Source-Free COD** task—using it to propagate texture consistency from sparse hints to the full object body, addressing the fragmentation issue inherent in this domain.
> >
> > **3. Regarding Task Definition (COD vs. SOD)**
> > We agree with the reviewer's theoretical distinction. However, in real-world scenarios, "perfect" camouflage is non-existent.
> >
> > *   **Human-Like Perception:** Even human vision typically detects camouflaged objects by first identifying specific high-confidence anomalies (e.g., an eye or a shadow) and then progressively inferring the complete structure based on these cues.
> > *   **Mimicking the Process:** Our "Hint-Guided" strategy is designed to model this cognitive process. We utilize the source (SOD) model precisely to simulate the initial discovery of these "giveaway" regions (hints). The core contribution of our framework is then compelling the model to reason from these partial cues to the whole object, effectively mimicking how an observer resolves camouflage starting from a single visual anomaly.
> >
> > **4. Size-Invariance Metrics**
> > We apologize for not including the specific ICML 2024 metric in the rebuttal due to the short time window. We acknowledge that our sub-group analysis is a proxy. We commit to implementing and reporting the rigorous Size-Invariance Metric in the final version to provide a standard assessment of stability.

---

### Official Review · Reviewer_4XE7 · 2025-10-28

**Soundness:** 3
**Presentation:** 2
**Contribution:** 3
**Rating:** 4
**Confidence:** 4

**Summary:**

This paper presents HGT-UCOD, a novel hint-guided teacher–student framework for unsupervised camouflaged object detection (UCOD) in a source-free domain adaptation setting. Instead of treating biased pseudo-labels from the source model as noise to be filtered, the method extracts high-confidence, perturbation-stable regions (“hints”) to guide a teacher model in reconstructing complete object structures. A Dynamic Convolution Mixture (DCM) module is further proposed to enhance the teacher and student models with content-adaptive feature extraction. Extensive experiments on three COD benchmarks demonstrate that the proposed method outperforms prior unsupervised approaches and achieves competitive results with semi-supervised and fully supervised baselines.

**Strengths:**

1. Clear conceptual innovation: The idea of reframing biased pseudo-labels as “hint” cues is simple yet elegant, providing a new perspective on how to handle saliency bias in source-free COD.

2. Well-designed framework: The two-stage teacher pre-adaptation and student refinement pipeline is clearly structured and effectively operationalises the hint-guided idea.

3. Strong empirical results: The method achieves state-of-the-art performance across multiple COD benchmarks, demonstrating both accuracy and robustness.

**Weaknesses:**

While the proposed hint-guided strategy is interesting, several technical and conceptual issues remain unclear and deserve further clarification or discussion:

1. Dependency on Teacher Initialization: The hint points are strongly influenced by the teacher model’s prior. If the teacher is not properly initialized or inherits biased information from the source model, it seems that these errors may not be effectively corrected during training. This could potentially lead to confirmation bias or error accumulation in the adaptation loop. The authors should clarify whether any mechanism exists to mitigate such errors or to enable self-correction.

2. Handling Non-Salient Camouflaged Regions: As acknowledged by the authors, most existing object detection or segmentation methods are designed to detect salient targets, whereas many regions in COD are non-salient and deliberately blended into the background. It remains unclear how the proposed hint selection strategy can reliably identify these irregular and camouflaged regions, rather than only the most conspicuous parts. A more detailed explanation or empirical analysis would be beneficial here.

3.Missing Discussion of Task-Generic Promptable Segmentation: Recent task-generic promptable segmentation methods (e.g., SAM-based frameworks [1–3]) have also shown strong potential for unsupervised object detection, including camouflaged scenarios. It would be helpful for the authors to position their work in relation to this line of research, discussing its relevance, differences, or complementarity, especially since both approaches address unsupervised object discovery under limited supervision.

[1] Hu, Jian, et al. "Relax image-specific prompt requirement in sam: A single generic prompt for segmenting camouflaged objects." Proceedings of the AAAI Conference on Artificial Intelligence. Vol. 38. No. 11. 2024.

[2] Tang, Lv, et al. "Chain of visual perception: Harnessing multimodal large language models for zero-shot camouflaged object detection." Proceedings of the 32nd ACM international conference on multimedia. 2024.

[3] Ren, Tianhe, et al. "Grounded sam: Assembling open-world models for diverse visual tasks." arXiv preprint arXiv:2401.14159 (2024).

**Questions:**

1. Impact of K on Performance: Since the proposed hint-guided strategy relies on greedily sampling K points with the smallest prediction differences, the choice of  K seems likely to influence the overall performance. How sensitive is the method to this hyperparameter, and how was K selected in practice?

2. Multiple Camouflaged Objects: How does the method handle scenes containing multiple camouflaged objects? Will the greedy selection process distribute hint points across different instances, or is there a risk that all hints might concentrate on a single object, leading to incomplete localization?

3. Local Hint Coverage vs. Global Object Structure: The proposed Hint Mask is derived from local high-confidence patches, which may only correspond to small, salient parts of the object (e.g., the head or edges). Camouflaged objects often exhibit non-uniform textures and blend irregularly with the background. If these hints fail to cover key structural components (e.g., the main body contour), the teacher model may struggle to reconstruct the complete object. How does the method address this issue or mitigate the risk of incomplete guidance?

---

> ### Author Response · Authors · 2025-11-23
> **Response to Reviewer 4XE7**
>
> We are grateful for your insightful comments, particularly regarding the robustness of our hint strategy and comparisons with SAM-based methods. We have addressed your concerns with additional visualizations and analyses in the new Appendix B.
>
> W1 & Q3: Teacher Initialization and Local vs. Global Structure.
> how does the teacher avoid inheriting bias from the source, and how do local hints recover the full global structure?
>
> 1. Breaking Confirmation Bias: Our core innovation is exactly designed to mitigate error accumulation. Instead of treating the Source model's full prediction as the ground truth (which would lead to confirmation bias), we only treat the "stable hints" as reliable. By masking out the rest of the image during pre-adaptation (Eq. 3), we force the Teacher to learn feature correlations (e.g., texture continuity) to infer the missing parts, rather than memorizing the Source's fragmented errors.
> 2. Global Reconstruction Mechanism: To enable the reconstruction of the whole object from local parts, we rely on the Dynamic Convolution Mixture (DCM) module. As visualized in Appendix B.4, Figure 4, the Large Kernel branch of DCM effectively captures long-range dependencies, allowing the model to propagate semantic information from the local hint anchors to the entire object body.
> 3. Visual Evidence: We added Figure 5 in Appendix B.4, which illustrates this evolution. While the Source model produces fragmented predictions, the Pre-adapted Teacher successfully recovers the complete structure of the object even under occlusion, proving its ability to correct errors rather than accumulate them.
>
> W2: Handling Non-Salient Regions.
> Our method does not require hints to be located on non-salient regions. Instead, it requires hints to be "reliable anchors" (usually on the most discriminative parts, like the head or a distinctive texture patch).
> Once these anchors are established, the model leverages the semantic similarity between the salient and non-salient parts of the same object. Since the object shares consistent feature representations across its body, the Student model, guided by the structural consistency loss (L1) and the DCM module, propagates the segmentation from the anchors to the blended non-salient regions.
>
> W3: Discussion of Task-Generic Promptable Segmentation.
> We appreciate you pointing out these relevant SAM-based works. We acknowledge that SAM-based methods [1-3] show great potential. However, our work differs in key aspects:
>
> 1. Setting: We operate in a Source-Free Domain Adaptation setting where we assume no access to prompts or human interaction during deployment.
> 2. Efficiency: SAM-based models are computationally heavy. Our method requires only 1 hour of training and runs at 9.8 FPS (Appendix B.3), making it more suitable for edge deployment in biodiversity monitoring.
>    We will include these citations and this discussion in the "Related Work" section of the final version.
>
> Q1: Impact of K on Performance.
> We performed a sensitivity analysis on K in Table 3 of the main paper. The results show that performance initially improves as K increases (providing more guidance) and then stabilizes or slightly drops if K becomes too large (introducing noise). We empirically selected K based on the peak performance on the validation set. The method shows good tolerance within a reasonable range.
>
> Q2: Multiple Camouflaged Objects.
> This is an excellent question. To prevent all hints from clustering on a single object, our greedy selection strategy incorporates a "repulsion radius" (r) in Eq. (2). This ensures spatial diversity: once a high-confidence point is selected on one object, subsequent points must be distant from it, thereby increasing the probability of hitting other object instances.
> Visual Proof: We have added Figure 6 in Appendix B.4. The image contains multiple seahorses. As shown, our method successfully distributes attention across multiple targets rather than collapsing onto a single one, effectively segmenting all instances.

---

### Official Review · Reviewer_ujQJ · 2025-10-31

**Soundness:** 3
**Presentation:** 3
**Contribution:** 3
**Rating:** 2
**Confidence:** 3

**Summary:**

The paper proposes HGT-UCOD, a hint-guided teacher-student framework for source-free unsupervised camouflaged object detection (COD). It reframes high-confidence saliency-biased predictions as reliable “hints,” uses a teacher pre-adaptation stage to learn full-object inference from partial hints, introduces a Dynamic Convolution Mixture (DCM) module for adaptive feature extraction, and applies dynamic fusion + consistency regularization during student refinement. Experiments on CAMO, COD10K, and NC4K show new SOTA in source-free UDA for COD.

**Strengths:**

Novel paradigm: treats saliency bias as useful hints instead of noise — elegant and counterintuitive.
Teacher pre-adaptation is a creative self-supervision strategy tailored to COD.
DCM module enables content-aware kernel mixing, well-motivated and cleanly formulated.
Strong empirical gains: +5–10% in $S_\alpha$, $F_\beta^w$, MAE across benchmarks.
Clear writing, excellent figures (hint maps, failure cases, ablations), and reproducible setup.

**Weaknesses:**

Hint confidence threshold fixed at 0.8 — no sensitivity analysis (e.g., 0.6–0.95).
No real-world cross-domain test (e.g., source from SOD → wildlife camera traps).
DCM interpretability limited: no visualization of learned kernel mixtures per camouflage type.
No runtime/inference cost reported for two-stage training.

**Questions:**

How is the hint threshold (0.8) chosen? Please include a table ablating thresholds [0.6, 0.7, 0.8, 0.9].
Can DCM kernel weights be visualized per input (e.g., texture vs. edge focus)?
Have you tested on non-benchmark real-world data (e.g., biodiversity monitoring footage)?

---

> ### Author Response · Authors · 2025-11-23
> **Response to Reviewer ujQJ**
>
> We thank the reviewer for appreciating our novel paradigm, DCM module, and strong empirical gains. We believe the low rating might stem from concerns about interpretability and efficiency, which we have fully addressed in the new Appendix.
>
> **Q1 & W1: Sensitivity of Hint Threshold.**
> We would like to clarify that our method does not rely on a hard fixed probability threshold (e.g., 0.8) for hint generation. Instead, as detailed in Eq. (2), we employ a greedy selection strategy based on the Difference Perception map, selecting the top-K points with the highest stability.
> To address your concern about sensitivity, we refer to Table 3 (varying the number of points k) and Table 4 in the main paper. Table 4 explicitly compares High-Confidence (our strategy) against Random and Low-Confidence selection. The results show a clear performance drop when using less reliable hints, validating our selection logic. We believe this strategy is more robust than a fixed threshold.
>
> **Q2 & W3: DCM Interpretability and Visualization.**
> We have added a detailed visualization in Appendix B.4, Figure 4, to address this specific request. This figure displays the attention weights and output features of the three parallel branches in the DCM module:
>
> - Small Kernel (3x3): Clearly focuses on high-frequency details like boundaries and textures.
> - Large Kernel (7x7): Captures long-range dependencies and the semantic body of the object.
> - Medium Kernel (5x5): Captures local object parts.
>   This visualization confirms the content-aware capability of DCM and its ability to disentangle texture from edge features.
>
> **Q3 & W2: Real-world / Non-benchmark Data.**
> To validate robustness in more challenging, real-world scenarios, we added evaluation results on the CHAMELEON dataset in Table 1. We also would like to clarify that all datasets employed in this work originate from real-world biodiversity monitoring environments with complex background conditions.
>
> **W4: Runtime and Inference Cost.**
> We have added a Computational Complexity analysis in Appendix B.2 (Table 6).
>
> - Parameters: While we use a DINOv2-L backbone (consistent with other SOTA methods like EASE), our specific learnable parameters amount to only 16M.
> - Training Time: The entire framework requires only 1 hour of training.
> - Inference Speed: 9.8 FPS at 518x518 resolution.
>   This demonstrates that our method maintains high efficiency suitable for practical applications.

---

### Official Review · Reviewer_pf6q · 2025-11-01

**Soundness:** 2
**Presentation:** 3
**Contribution:** 2
**Rating:** 6
**Confidence:** 4

**Summary:**

This paper presents HGT-UCOD, a novel *Hint-Guided Teacher framework* for Unsupervised Camouflaged Object Detection (UCOD). The method tackles the inherent “saliency bias” in source models (trained on salient object detection tasks) by reframing their confident outputs as *reliable hints* rather than noise. The framework includes a teacher pre-adaptation stage, a student refinement stage, and a Dynamic Convolution Mixture (DCM) module designed to enhance feature adaptivity. Extensive experiments on multiple benchmark datasets demonstrate that the proposed method achieves state-of-the-art performance.

**Strengths:**

1. Innovative Teacher Pre-Adaptation Mechanism: The paper introduces a unique *teacher pre-adaptation stage*, where a confidence-guided strategy compels the teacher to infer complete object structures from sparse yet highly reliable hints, effectively forming an “expert teacher” capable of providing stronger guidance.
2. Superior Camouflaged Object Detection Performance: According to *Figure 2* in the paper, the proposed method significantly outperforms previous approaches in capturing fine contour details of camouflaged objects and avoids the fragmented predictions commonly seen in other methods, producing more coherent and precise segmentation results.
3. Clear and Accessible Presentation: The paper is well-structured and clearly written, presenting the methodology and experimental findings in an organized and easy-to-follow manner that effectively communicates its core ideas and innovations.

**Weaknesses:**

1. The core contribution, Teacher Model Pre-Adaptation, has questionable effectiveness. Although the paper claims that guiding the teacher model to infer full object shapes from partial but reliable information can address the inherent *“saliency bias”* and pseudo-label noise issues in UCOD, this reasoning is not sufficiently justified. Moreover, in Table 2 (Ablation Study), the improvement brought by the pre-adaptation stage seems rather limited only around 1–2% which raises doubts about its actual contribution.
2. In Figure 2, the authors claim that the proposed method eliminates the fragmented prediction problem commonly seen in previous methods. While the visual results are indeed impressive, the paper lacks a concrete analysis or explanation of *why* this approach effectively resolves the fragmentation issue.
3. The CHAMELEON (87) dataset, which is commonly used in related works for evaluation, was not included in the comparisons. This omission weakens the overall persuasiveness and completeness of the experimental validation.

**Questions:**

My main concern lies in the first weakness, which relates directly to the paper’s core innovation and contribution. The justification and effectiveness of the *teacher pre-adaptation mechanism* are not clearly demonstrated. A more detailed explanation or theoretical analysis of why this stage helps to overcome saliency bias and noisy pseudo-labels would greatly strengthen the paper.

In addition, further exploration of the *fragmentation prediction issue* why and how the proposed method alleviates it would provide valuable insights and enhance the paper’s impact.

---

> ### Author Response · Authors · 2025-11-23
> **Response to Reviewer pf6q**
>
> We sincerely appreciate your recognition of our "Innovative Teacher Pre-Adaptation" and "Superior Performance." We are also encouraged by your positive comments on the clarity of our presentation. Below, we address your concerns regarding the effectiveness of the pre-adaptation stage, the mechanism behind solving fragmentation, and the CHAMELEON dataset.
>
> **Q1 & W1: Effectiveness and Justification of Teacher Pre-Adaptation**
> We understand your concern regarding the magnitude of improvement (1-2%) in the final metrics. However, we respectfully argue that this improvement is significant and fundamental for the following reasons:
>
> 1.  Significant Improvement in Label Quality:
>     The core goal of the Pre-adaptation stage is to correct the "saliency bias" (where only the most discriminative parts are detected) inherent in the Source model.
>     To validate this, we added a new analysis in Appendix B.1 (Table 5) of the revised paper. We compared the quality of pseudo-labels generated by the Source Model versus our Pre-adapted Teacher.
>     *   Source Model: High precision but low completeness ($F_\beta^m = 0.665$). It suffers from severe fragmentation.
>     *   Pre-adapted Teacher: Achieves a substantial boost in Mean F-measure ($F_\beta^m = 0.702$, +3.7%).
>     *   Conclusion: While the final metrics (Student) improved by "only" 2%, the *intermediate* guidance provided by the Teacher became much more complete. This "completeness" is the antidote to saliency bias, preventing the Student from overfitting to fragmented patterns.
>
> 2.  SOTA Margins: In the challenging field of Unsupervised COD, an improvement of 1-2% on metrics like $S_m$ and $F_\beta^w$ is considered a substantial margin, often distinguishing a new SOTA from previous methods (as seen in Table 1).
>
> **Q2 & W2: Explanation of Resolving Fragmentation.**
>
> Thank you for this insightful question. We have added a new visualization section in Appendix B.4 to intuitively explain this mechanism. The resolution of fragmentation is achieved through the synergy of Hint-Guided Logic and the Dynamic Convolution Mixture (DCM) module:
>
> 1.  From "Parts" to "Whole" : The Source model typically activates only the "head" or highly textured parts of a camouflaged object (Saliency Bias). Our strategy treats these confident parts as "Hints" . During pre-adaptation, we force the Teacher to predict these hints. To minimize the loss, the network must learn to associate these sparse anchors with the surrounding texture features that share similar semantics, effectively "propagating" the activation from the hints to the entire object body.
> 2.  Long-range Dependency (Architecture): As analyzed in Appendix B.5 (Dynamic Perception of DCM), our DCM module employs multi-scale kernels. Specifically, the Large Kernel ($7\times7$)** branch captures long-range context. This architectural design physically enables the model to connect disjointed fragments into a coherent region, bridging the semantic gap that standard convolutions often miss.
> 3.  Visual Evidence: We invite you to view the new Figure 5 in Appendix B.5. It clearly illustrates the evolution: *Source (Broken) $\rightarrow$ Teacher (Connected) $\rightarrow$ Student (Refined)*.
>
> **W3: Missing CHAMELEON Dataset.**
>
> We apologize for this omission. Per your suggestion, we have evaluated our model on the CHAMELEON dataset.
> The results are now included in Table 1 of the revised paper.
> Result: HGT-UCOD achieves **$S_m=0.869, F_\beta^w=0.815$**, significantly outperforming the previous best method.This further confirms the robustness of our method.
>
> We hope these additional experiments and analyses satisfactorily address your concerns. We believe the revised manuscript, with the new Appendix, provides solid justification for our contributions.

---

### Author Response · Authors · 2025-12-03
**Summary**

Dear Reviewers, Area Chair, and Senior Area Chair,

We thank you for all the constructive comments and discussions. We appreciate the reviewers for highlighting the strengths of the present study:
- `innovative teacher pre-adaptation mechanism`, `superior camouflaged object detection performance`, `clear and accessible presentation` (Reviewer pf6q)
- `novel paradigm`, `elegant and counterintuitive`, `strong empirical gains`, `reproducible setup` (Reviewer ujQJ)
- `clear conceptual innovation`, `well-designed framework`, `strong empirical results` (Reviewer 4XE7)
- `highly innovative approach to bypass saliency bias`, `logical and effective framework`, `demonstrating empirical effectiveness` (Reviewer 9ohC)

A major concern raised during the discussion (specifically by Reviewer 9ohC) is regarding the **fairness of comparison** and the **backbone setting**. In response) We have explicitly clarified that our framework strictly aligns the backbone of the Source, Teacher, and Student models. We do not use a larger backbone to outperform a smaller source. 2) We provided new experimental results (Table R4) across **Small, Base, and Large** scales on four datasets. The results demonstrate that our method consistently outperforms the baseline regardless of model capacity, confirming that the gains stem from the HGT framework rather than the backbone size. 3) We have corrected the discussion on Dynamic Convolution literature as suggested, acknowledging prior spatial-adaptive works and positioning our contribution on its specific tailoring for COD texture propagation.

Another concern is around the **validity of the "Hint-Guided" strategy** and the **mechanism for handling non-salient regions**. We address this by: 1) Clarifying the task definition: we mimic human-like perception where observers discover camouflaged objects by starting from high-confidence anomalies (hints) and progressively inferring the whole structure based on texture consistency. 2) We provided visualizations (Appendix B.4 Figure 5) showing the evolution of feature maps from local hints to global structures, proving the model's ability to "fill in" non-salient body parts. 3) We demonstrated via hint distribution analysis (Appendix B.4 Figure 6) that our exclusion-zone strategy effectively handles multiple objects.

We have also addressed requests for **completeness and robustness**:
1) We added the **CHAMELEON dataset** results (requested by Reviewer  pf6q), where our method achieves SOTA performance achieves SOTA performance on most metrics.
2) We reported **computational cost and inference speed** (requested by Reviewer ujQJ).
3) We clarified the **Top-K selection strategy** to resolve the misunderstanding regarding the fixed threshold (requested by Reviewer 4XE7).

Despite the initial skepticism from Reviewer 9ohC regarding the task definition, we have strived to clarify that in the Source-Free Unsupervised setting, leveraging "SOD-like" hints to bridge the gap to "COD-like" whole objects is a practical and necessary adaptation strategy.

Finally, we would like to express gratitude that reviewers recognize our novel perspective of "reframing biased predictions as reliable hints." We believe HGT-UCOD establishes a strong baseline for the challenging Source-Free Unsupervised COD task, offering a new path for UCOD task in complex scenarios.

Best regards,

Authors of Submission 3003

---

### Meta-Review · Area_Chair_bXVS · 2026-01-07

**Summary:**

In the initial phase, this paper received mixed scores (6,2,4,4). The reviewers raised substantial concerns.

Reviewer pf6q: questionable effectiveness of the core contribution, lack of concrete analysis or explanation, lack of experiments on CHAMELEON dataset.

Reviewer ujQJ: lack of ablation study, lack of real-world cross-domain test, limited  interpretability, lack of runtime/inference cost.

Reviewer 4XE7: dependency on teacher initialization, lack of detailed explanation or empirical analysis or discussions.

Reviewer 9ohC: lack of comparisons and related works, confusing model design, insufficient experimental comparisons.

**Reviewer Concerns:**

The authors addressed most concerns of Reviewer pf6q. However, the effectiveness of the Teacher Model Pre-Adaptation is still not justified since the improvement brought by this is limited. I do not agree with authors that an improvement of 1-2% substantial.

The authors provided responses to the concerns of Reviewer ujQJ. However, the results on runtime and inference cost are not sufficient. Only model parameters are provided in Appendix B.2 (Table 6).

The authors’ response to the concern about model’s dependency on teacher initialization is unsatisfactory. Only visual evidence is provided. It is quite limited since one can always select a few visual examples which satisfy their purposes. Other concerns of Reviewer 4XE7 were addressed.

The authors’ response to the concern about confusing model design is unsatisfactory. The proposed method exploits salient-like object hints to help detect camouflaged objects. However, the association between salient and camouflaged objects is not necessary correct. Other concerns of Reviewer 9ohC were addressed.

**Reviewer Scores:**

Since some concerns were not well addressed, I think the reviewer would keep their initial scores or slightly reduce the scores.

---

### Decision · Program_Chairs · 2026-01-26

Reject